# PROMPTING LANGUAGE-INFORMED DISTRIBUTION FOR COMPOSITIONAL ZERO-SHOT LEARNING

## ABSTRACT

Compositional zero-shot learning (CZSL) task aims to recognize unseen compositional visual concepts, *e.g.*, `sliced tomatoes`, where the model is learned only from the seen compositions, *e.g.*, `sliced potatoes` and `red tomatoes`. Thanks to the prompt tuning on large pre-trained visual language models such as CLIP, recent literature shows impressively better CZSL performance than traditional vision-based methods. However, the key aspects that impact the generalization to unseen compositions, including the *diversity* and *informativeness* of class context, and the *entanglement* between visual primitives, *i.e.*, state and object, are not properly addressed in existing CLIP-based CZSL literature. In this paper, we propose a model by prompting the language-informed distribution, aka., $\mathbb{PLID}$, for the CZSL task. Specifically, the $\mathbb{PLID}$ leverages pre-trained large language models (LLM) to 1) formulate the language-informed class distributions which are diverse and informative, and 2) enhance the compositionality of the class embedding. Moreover, a visual-language primitive decomposition (VLPD) module and a stochastic logit mixup (SLM) strategy are proposed to dynamically fuse the decisions from the compositional and the primitive logit space. Orthogonal to the existing literature of soft, hard, or distributional prompts, our method advocates prompting the LLM-supported class distribution that leads to a better zero-shot generalization. Experimental results on MIT-States, UT-Zappos, and C-GQA datasets show the superior performance of the $\mathbb{PLID}$ to the prior arts. The code and models will be publicly released.

## 1 INTRODUCTION

Compositional visual recognition is a fundamental characteristic of human intelligence (Lake et al., 2017) but it is challenging for modern deep learning systems. For example, humans can easily recognize unseen `sliced tomatoes` after seeing `sliced potatoes` and `red tomatoes`. Such a compositional zero-shot learning (CZSL) capability is valuable in that, novel visual concepts from a huge combinatorial semantic space could be recognized without "seeing" any of their training data. For example, C-GQA (Naeem et al., 2021) dataset contains 413 states and 674 objects. This implies a total of at least 278K compositional classes in an open world while only 2% of them are accessible in training. Therefore, CZSL can significantly reduce the need for large-scale training data.

Traditional vision-based methods either directly learn the visual feature of compositions, or try to first decompose the visual data into representations of simple primitives, *i.e.*, states and objects, and then learn to re-compose the compositions (Misra et al., 2017; Atzmon et al., 2020; Zou et al., 2020; Huynh & Elhamifar, 2020; Karthik et al., 2022; Tokmakov et al., 2019; Naeem et al., 2021; Zhang et al., 2022b; Mancini et al., 2021; Li et al., 2022). Thanks to the recent large pre-trained vision-language models (VLM) such as CLIP (Radford et al., 2021), recent state-of-the-art CZSL methods have been developed (Nayak et al., 2023; Lu et al., 2023; Xu et al., 2022; Huang et al., 2023). For instance, CSP (Nayak et al., 2023) inherits the hard prompt template of the CLIP, *i.e.*, *a photo of* [`state`][`object`] where only the embeddings of the state-object pairs are trained. The following methods (Lu et al., 2023; Xu et al., 2022; Huang et al., 2023) use soft prompt introduced in CoOp (Zhou et al., 2022b), where the embeddings of the prompt template are jointly optimized, leading to a better CZSL performance. The impressive performance of CLIP-based CZSL methods benefits from the sufficiently good feature alignment between the image and text modalities, and the prompting techniques for adapting the aligned features to recognizing compositional classes.

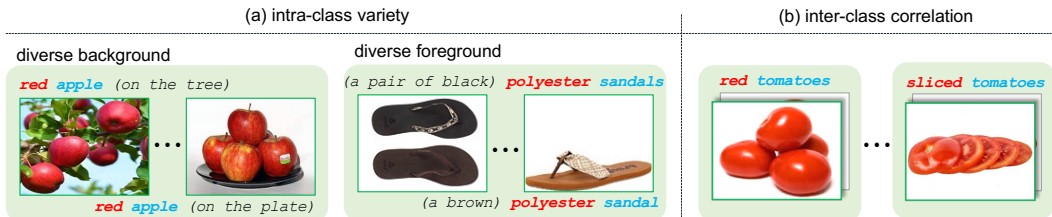

Figure 1: **Challenges of compositional recognition.** **(a)** images of the same compositional class appear differently due to diverse visual backgrounds or foregrounds. **(b)** `red tomatoes` and `sliced tomatoes` are visually correlated because 1) both are `tomatoes` object, and 2) the object `tomatoes` is inherently entangled with the state `red`, resulting in the need of primitive decomposition.

Despite the success of existing CLIP-based methods, we find several key considerations to prompt the pre-trained CLIP for better CZSL modeling. First, the *diversity* and *informativeness* of prompts are both important to distinguish between compositional classes. CZSL can be treated as zero-shot learning on fine-grained categories, which requires a fine-grained context to prompt the CLIP model (Radford et al., 2021; Lu et al., 2022). However, to contextualize a class with fine granularity, the hard prompt in Radford et al. (2021) suffers from the heuristic design of prompt templates, and a single prompt for each class lacks diversity to capture the intra-class variance of visual data (Fig. 1a). Though the ProDA (Lu et al., 2022) proposes to learn a collection of prompts that formulate class-specific distribution to address the diversity, the lack of *language informativeness* in their prompts limits their performance on fine-grained compositional categories. Second, the entanglement between visual primitives, *e.g.* red and `tomatoes` in Fig. 1b, incurs difficulty in learning decomposable visual representations that are useful for compositional generalization (Liu et al., 2022; Karthik et al., 2022), while such a capability is missing in (Nayak et al., 2023; Xu et al., 2022). Though the more recent work (Lu et al., 2023; Huang et al., 2023) learn to decompose the primitives and considers the re-composed compositional predictions, their language-only decomposition and probability-level mixup potentially limit the generalizability in the open-world.

In this paper, we propose a novel CLIP-based method for the CZSL task by prompting the language-informed distributions ($\mathbb{PLID}$) over both the compositional and primitive categories. To learn the diverse and informative textual class representations, the $\mathbb{PLID}$ leverages off-the-shelf large language models (LLM) to build the class-specific distributions and to enhance the class embeddings. Furthermore, we propose a visual language primitive decomposition (VLPD) module to decompose the image data into simple primitives. Eventually, the compositional classification is enhanced by our stochastic logit mixup (SLM), which takes the merits of both the compositional and primitive recognitions. The proposed $\mathbb{PLID}$ shows state-of-the-art performance on CZSL benchmarks such as MIT-States (Isola et al., 2015), UT-Zappos (Yu & Grauman, 2014), and C-GQA (Naeem et al., 2021).

Note that our method is orthogonal to the existing hard prompt (Radford et al., 2021), soft prompt tuning (Zhou et al., 2022b), and prompt distribution learning (Lu et al., 2022; Kwon et al., 2023; Liu et al., 2023; Derakhshani et al., 2023). We advocate prompting the distribution of informative LLM-based class descriptions. From a classification perspective, this is grounded on the classification-by-description (Menon & Vondrick, 2023; Maniparambil et al., 2023; Yan et al., 2023; He et al., 2023), that LLM-generated text enables more informative class representations. Compared to the deterministic soft/hard prompt aforementioned, our distribution modeling could capture the intra-class diversity for better zero-shot generalization. Compared to the existing prompt distribution learning approaches, the class context is more linguistically interpretable and provides fine-grained descriptive information about the class. Our method is also parameter-efficient without the need to optimize a large collection of prompts. Specific to the CZSL task, the enhanced class embeddings by LLM descriptions enable visual language primitive decomposition and decision fusion in both compositional and primitive space, which eventually benefits the generalization to the unseen.

In summary, the contributions are as follows. a) We develop a $\mathbb{PLID}$ method that advocates prompting the language-informed distribution for compositional zero-shot learning, which is orthogonal to existing soft/hard and distributional prompt learning. b) We propose primitive decomposition and stochastic logit mixup to fuse the classification decision from compositional and primitive predictions. c) We empirically show that $\mathbb{PLID}$ could achieve superior performance to prior arts in both the closed-world and open-world settings on MIT-States, UT-Zappos, and C-GQA datasets.

## 2 RELATED WORK

**Prompt Learning in VLM** Vision-Language Models (VLM) such as the CLIP (Radford et al., 2021) pre-trained on web-scale datasets recently gained substantial attention for their strong zero-shot recognition capability on various downstream tasks. Such a capability is typically achieved by performing prompt engineering to adapt pre-trained VLMs. Early prompting technique such as the hard prompt in CLIP uses the heuristic template "*a photo of* [CLS]" as the textual input. Recently, the soft prompt tuning method in CoOp (Zhou et al., 2022b), CoCoOp (Zhou et al., 2022a), and ResPT (Razdaibiedina et al., 2023) that uses learnable embedding as the textual context of class names significantly improved the model adaptation performance. This technique is further utilized in MaPLe (Khattak et al., 2023) that enables multi-modal prompt learning for both image and text. However, the prompts of these methods are deterministic and lack the diversity to capture the appearance variety in fine-grained visual data, so they are prone to overfitting the training data. To handle this issue, ProDA (Lu et al., 2022) explicitly introduces a collection of soft prompts to construct the class-specific Gaussian distribution, which results in better zero-shot performance and inspires the recent success of PPL (Kwon et al., 2023) in the dense prediction task. Similarly, the PBPrompt (Liu et al., 2023) uses neural networks to predict the class-specific prompt distribution and utilizes optimal transport to align the stochastically sampled soft prompts and image patch tokens. The recent work (Derakhshani et al., 2023) assumes the latent embedding of prompt input follows a Gaussian prior and adopts variational inference to learn the latent distribution. In this paper, in order to take the merits of the *informativeness* of hard prompt and the *diversity* of distributional modeling, we adopt the soft prompt to adapt the distributions supported by LLM-generated class descriptions.

**Compositional Zero-Shot Learning (CZSL)** For a long period, the CZSL task has been studied from a vision-based perspective in literature. They either directly learn the compositional visual features or disentangle the visual features into simple primitives, *i.e.*, states and objects. For example, (Nagarajan & Grauman, 2018; Li et al., 2020; Naeem et al., 2021) performs a direct classification by projecting the compositional visual features into a common feature space, and (Lu et al., 2016; Misra et al., 2017; Atzmon et al., 2020; Huynh & Elhamifar, 2020; Zou et al., 2020; Karthik et al., 2022; Liu et al., 2022) decompose the visual feature into simple primitives so that the compositional recognition can be achieved by learning to recompose from the primitives. Though the recent large-scale pre-trained CLIP model shows impressive zero-shot capability, it is found to struggle to work well for compositional reasoning (Ma et al., 2023; Yuksekgonul et al., 2023; Lewis et al., 2022). Thanks to the recent prompt learning (Zhou et al., 2022b), the CZSL task has been dominated by CLIP-based approaches (Nayak et al., 2023; Lu et al., 2023; Xu et al., 2022; Huang et al., 2023). The common idea is to prompt the frozen CLIP model to separately learn the textual embeddings of simple primitives, which empirically show strong compositionality for zero-shot generalization. However, these methods tend to overfitting due to the lack of prompt diversity or language informativeness. In this paper, based on the frozen CLIP, we leverage LLMs to enhance the compositionality of text embeddings and propose to decompose both the image and text modalities for better compositional recognition in an open world.

## 3 PRELIMINARIES

**CZSL Task Formulation** The CZSL task aims to recognize images of a compositional category $y \in \mathcal{C}$, where the semantic space $\mathcal{C}$ is a Cartesian product between the state space $\mathcal{S} = \{s_1, \ldots, s_{|\mathcal{S}|}\}$ and object space $\mathcal{O} = \{o_1, \ldots, o_{|\mathcal{O}|}\}$, *i.e.*, $\mathcal{C} = \mathcal{S} \times \mathcal{O}$. For example, as shown in Fig. 1, a model trained on images of red apple and sliced tomatoes needs to additionally recognize an image of sliced apple. In training, only a set of **seen** compositions is available. In closed-world testing, the model needs to recognize images from both the **seen** compositions in $\mathcal{C}^{(s)}$ and the **unseen** compositions in $\mathcal{C}^{(u)}$ that are assumed to be feasible, where the cardinality $|\mathcal{C}^{(s)} \cup \mathcal{C}^{(u)}| \ll |\mathcal{C}|$ since most of the compositions in $\mathcal{C}$ are practically not feasible. In open-world testing, the model needs to recognize images given any composition in $\mathcal{C}$.

**VLMs for CZSL** Large pre-trained VLMs such as CLIP (Radford et al., 2021) have recently been utilized by CSP (Nayak et al., 2023) for the CZSL task. The core idea of CSP is to represent the text embeddings of states in $\mathcal{S}$ and objects in $\mathcal{O}$ as learnable parameters and contextualize them with the hard prompt template "*a photo of* [s][o]" as the input of the CLIP text encoder, where [s] $\in \mathcal{S}$ and

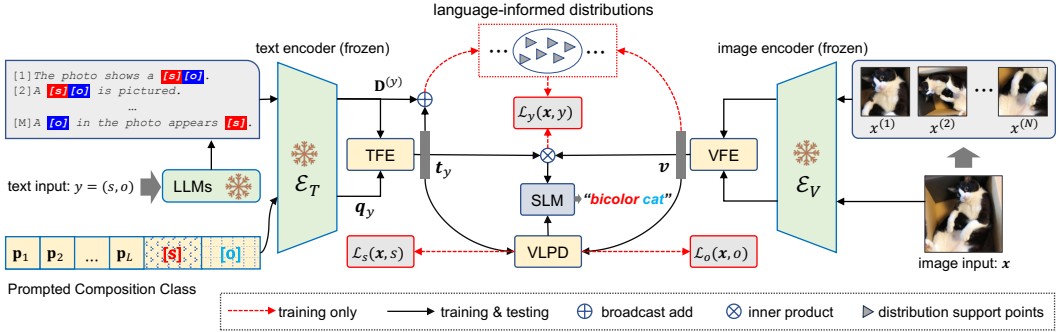

Figure 2: Overview of $\mathbb{PLID}$. The CZSL task is formulated to align the feature of image $\mathbf{x}$ with the learnable text features of compositional class $y = (s, o)$ based on frozen CLIP ($\mathcal{E}_T$ and $\mathcal{E}_V$). We propose the language-informed distributions (**LID**) which are constructed by the LLM-generated class descriptions and the soft prompts $\mathbf{p}_{1:L}$ for each state-object pair $(s, o)$. The features of the image and text are enhanced by text and visual feature enhancement (TFE and VEF). Furthermore, we propose the visual language primitive decomposition (**VLPD**) module to recompose the compositional logits, which are further fused with the compositional logit between $\mathbf{t}_y$ and $\mathbf{v}$ by our stochastic logit mix-up (**SLM**). With the compositional and primitive recognition, our model is jointly trained by loss functions $\mathcal{L}_y(\mathbf{x}, y)$, $\mathcal{L}_s(\mathbf{x}, s)$, and $\mathcal{L}_o(\mathbf{x}, o)$.

$[\mathsf{o}] \in \mathcal{O}$. Given an image $\mathbf{x}$, by using the cosine similarity ($\cos$) as the logit, the class probability of the composition $y$ is defined as $p_{\boldsymbol{\theta}}(y|\mathbf{x}) = \mathtt{softmax}(\cos(\mathbf{v}, \mathbf{t}_y))$, where $\boldsymbol{\theta}$ are the $|\mathcal{S}| + |\mathcal{O}|$ learnable parameters, $\mathbf{v}$ and $\mathbf{t}_y$ are the image feature and class text embedding, respectively.

In training, the prediction $p_{\boldsymbol{\theta}}(\hat{y}|\mathbf{x})$ is supervised by multi-class cross-entropy loss. In CZSL testing, a test image is recognized by finding the compositional class $c \in \mathcal{C}$ which has the maximum $\cos(\mathbf{v}, \mathbf{t}_c)$. The CSP method is simple, parameter-efficient, and largely outperforms traditional approaches. However, due to the lack of diversity and informativeness in prompting, the zero-shot capability of CLIP is not fully exploited by CSP for the CZSL task.

## 4 PROPOSED METHOD

**Overview** Fig. 2 shows an overview of the $\mathbb{PLID}$. The basic idea is to use LLMs to generate sentence-level descriptions for each compositional class, and learn to prompt the class-wise text distributions (supported by the descriptions) to be aligned with image data. Besides, we introduce visual language primitive decomposition (VLPD) and stochastic logit mixup (SLM) to enable recognition at both compositional and primitive levels. In testing, an image is recognized by fusing the decisions from the directly predicted and the recomposed compositions.

### 4.1 PROMPTING LANGUAGE-INFORMED DISTRIBUTION

**Motivation** To adapt the large pre-trained CLIP (Radford et al., 2021) to downstream tasks, recent distributional prompt learning (Lu et al., 2022; Kwon et al., 2023; Liu et al., 2023; Derakhshani et al., 2023) shows the importance of *context diversity* by distribution modeling for strong generalization. Motivated by the inherent fine-granularity of compositional recognition in the CZSL task, we argue that not only the context diversity but also the *context informativeness* by language modeling, are both important factors to adapt CLIP to the zero-shot learning task. The insight behind this is that the sentence-level descriptions could contextualize compositional classes in a more fine-grained manner than the prior arts. Therefore, we propose to address the two factors by learning to **P**rompt the **L**anguage-**I**nformed **D**istributions ($\mathbb{PLID}$) for the CZSL task.

**Compositional Class Description** To generate diverse and informative text descriptions for each compositional class, we adopt a similar way as (Menon & Vondrick, 2023) by prompting an LLM that shows instruction-following capability. An example below shows the format of the LLM instruction.

```
Keywords: sliced, potato, picture
Output: The picture features a beautifully arranged plate of thinly
    sliced potatoes.
###
```

See the Appendix B for more details. For each composition $y = (s, o)$, we generate $M$ descriptions denoted as $S^{(y)} = \{S_1^{(y)}, \ldots, S_M^{(y)}\}$ where $S_m^{(y)}$ is a linguistically complete sentence. Different to (Menon & Vondrick, 2023) that aims to interpret the zero-shot recognition by attribute phrases from LLMs, we utilize the LLM-based sentence-level descriptions in the CZSL task for two benefits: 1) provide diverse and informative textual context for modeling the class distributions that capture the intra-class variance, and 2) enhance the class embedding with fine-grained descriptive information.

**Language-Informed Distribution (LID)** For both the image and text modalities, we use the frozen CLIP model and learnable feature enhancement modules to represent the visual and language features, which are also adopted in existing CZSL literature (Lu et al., 2023; Huang et al., 2023).

Specifically, for the text modality, each composition $y$ is tokenized and embedded by CLIP embedding layer and further prompted by concatenating with learnable context vectors, *i.e.*, "$[\mathbf{p}_1] \ldots [\mathbf{p}_L][\mathbf{s}][\mathbf{o}]$", where $\mathbf{p}_{1:L}$ is initialized by "a photo of" and shared with all classes. Followed by the frozen CLIP text encoder $\mathcal{E}_T$, the embedding of class $y$ is $\mathbf{q}_y = \mathcal{E}_T([\mathbf{p}_1] \ldots [\mathbf{p}_L][\mathbf{s}][\mathbf{o}])$ where $\mathbf{q}_y \in \mathbb{R}^d$. Following the CZSL literature (Xu et al., 2022; Lu et al., 2023), here the soft prompt $\mathbf{p}_{1:L}$ and primitive embeddings $[\mathbf{s}][\mathbf{o}]$ are learnable while $\mathcal{E}_T$ is frozen in training.

To simultaneously address the lack of diversity and informativeness of the soft prompts, we propose to formulate the class-specific distributions supported by the texts $S^{(y)}$ and learn to prompt these distributions. Specifically, we encode $S^{(y)}$ by the frozen CLIP text encoder: $\mathbf{D}^{(y)} = \mathcal{E}_T(S^{(y)})$, where $\mathbf{D}^{(y)} \in \mathbb{R}^{M \times d}$. Then, we use $\mathbf{D}^{(y)}$ to enhance $\mathbf{q}_y$ by $\mathbf{t}_y = \Psi_{\text{TFE}}(\mathbf{q}_y, \mathbf{D}^{(y)})$ where $\Psi_{\text{TFE}}$ is the text feature enhancement (**TFE**) implemented by cross attention (Vaswani et al., 2017). Similarly, given an image $\mathbf{x}$, to mitigate the loss of fine-grained cues, we augment it with $N$ views to be $\mathbf{X} = \{\mathbf{x}^{(1)}, \ldots, \mathbf{x}^{(N)}\}$. Followed by the frozen CLIP visual encoder $\mathcal{E}_V$, the feature of $\mathbf{x}$ is enhanced by $\mathbf{v} = \Psi_{\text{VFE}}(\mathcal{E}_V(\mathbf{x}), \mathcal{E}_V(\mathbf{X}))$ where $\Psi_{\text{VFE}}$ is the visual feature enhancement (**VFE**) by cross attention.

We treat the enhanced text feature $\mathbf{t}_y$ of class $y$ as the class mean and $\mathbf{t}_y + \mathbf{D}^{(y)}$ as the distribution support points (**DSP**) that follow the Gaussian $\mathcal{N}(\mathbf{t}_y, \boldsymbol{\Sigma}_y)$. The motivation of $\mathbf{t}_y + \mathbf{D}^{(y)}$ is to enable the flexibility of DSP to traverse around in the $d$ dimensional space in training since $\mathbf{t}_y$ is trainable while $\mathbf{D}^{(y)}$ are pre-trained. For all $|\mathcal{C}^{(s)}|$ (denoted as $C$) seen compositional classes, we build joint Gaussian distributions $\mathcal{N}(\boldsymbol{\mu}_{1:C}, \boldsymbol{\Sigma}_{1:C})$ similar to ProDA (Lu et al., 2022), where the means $\boldsymbol{\mu}_{1:C} \in \mathbb{R}^{C \times d}$ are given by $\mathbf{t}_y$ over $C$ classes, and the covariance $\boldsymbol{\Sigma}_{1:C} \in \mathbb{R}^{d \times C \times C}$ is defined across $C$ classes for each feature dimension from DSP.

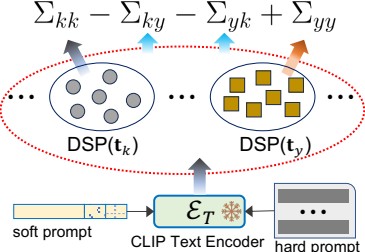

Figure 3: Hybrid prompting for intra- and inter-class covariance optimization.

*Remark*: Compared to the ProDA (Lu et al., 2022) that learns a collection of non-informative prompts, our DSPs are language-informed by $\mathbf{D}^{(y)}$ that provides more fine-grained descriptive information to help recognition and decomposition. Besides, our method is more parameter-efficient than ProDA since we only have a single soft prompt to learn. This is especially important for the CZSL task where there is a huge number of compositional classes. Lastly, we highlight the benefit of performing the intra- and inter-class covariance optimization induced by the learning objective of distribution modeling, which will be introduced below.

**Learning Objective** Given the visual feature $\mathbf{v} \in \mathbb{R}^d$ of image $\mathbf{x}$ and the text embeddings $\mathbf{t}_{1:C}$ from class-wise joint distributions $\mathcal{N}(\boldsymbol{\mu}_{1:C}, \boldsymbol{\Sigma}_{1:C})$, according to the (Lu et al., 2022), minimizing the cross-entropy loss is equivalent to minimizing the upper bound of negative log-likelihood (NLL):

$$\text{NLL}(x, y) = -\log \mathbb{E}_{\mathbf{t}_{1:C}} p(y|\mathbf{v}, \mathbf{t}_{1:C}) \leq -\log \frac{\exp(h_y/\tau)}{\sum_{k=1}^{C} \exp((h_k + h_{k,y}^{(m)})/\tau)} := \mathcal{L}_y(\mathbf{x}, y), \quad (1)$$

where the compositional logit $h_y = \cos(\mathbf{v}, \mathbf{t}_y)$, the pairwise margin $h_{k,y}^{(m)} = \mathbf{v}^\top \mathbf{A}_{k,y} \mathbf{v}/(2\tau)$ and $\mathbf{A} \in \mathbb{R}^{d \times C \times C}$ is given by $\mathbf{A}_{k,y} = \boldsymbol{\Sigma}_{kk} + \boldsymbol{\Sigma}_{yy} - \boldsymbol{\Sigma}_{ky} - \boldsymbol{\Sigma}_{yk}$. The covariance $\mathbf{A}_{k,y}$ indicates the correlation between the $k$-th out of $C$ classes and the target class $y$ on each of $d$ feature dimensions. The insight of minimizing $\mathcal{L}_y(\mathbf{x}, y)$ is illustrated in Fig. 3, which encourages minimizing intra-class variance by $\boldsymbol{\Sigma}_{yy}$ and $\boldsymbol{\Sigma}_{kk}$, and maximizing inter-class separability indicated by $\boldsymbol{\Sigma}_{ky}$ and $\boldsymbol{\Sigma}_{yk}$. In Appendix C, we discuss our workaround by covariance sharing when $C$ is too large to compute $\mathbf{A}$.

## 4.2 PRIMITIVES DECOMPOSITION AND DECISION FUSION

**Motivation**  Considering the fundamental challenge in the CZSL task, that the visual primitives are inherently entangled in an image, an unseen composition in testing can be hardly identified if its object (or its state) embedding is overfitted to the visual data of seen compositions. To this end, it is better to inherit the benefits of the decompose-recompose paradigm (Zou et al., 2020; Karthik et al., 2022; Liu et al., 2022) by decomposing visual features into simple primitives, *i.e.*, states and objects, from which the recomposed decision can be leveraged for zero-shot recognition. Thanks to the compositionality of CLIP (Wolff et al., 2023; Trager et al., 2023), such motivation can be achieved by the visual-language primitive decomposition (**VLPD**). See Fig. 4 and we explain it below. Based on VLPD, we propose the stochastic logit mixup to fuse the directly learned compositions and the recomposed ones.

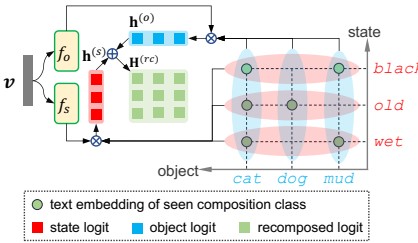

Figure 4: VLPD for recomposing.

**VLPD**  Specifically, we use two parallel neural networks $f_s$ and $f_o$ to decompose $\mathbf{v}$ into the state visual feature $f_s(\mathbf{v})$ and object visual feature $f_o(\mathbf{v})$, respectively, under the supervision of text features. To get the supervision, we group $\mathbf{t}_y$ over the subset $\mathcal{Y}_o$, in which all compositions share the same given object $o$ (see vertical ellipses in Fig. 4), and group $\mathbf{t}_y$ over the subset $\mathcal{Y}_s$, in which all compositions share the same given state $s$ (see horizontal ellipses in Fig. 4). Thus, given a state $s$ and an object $o$, the predicted object logit $h_s$ and state logit $h_o$ are computed by

$$h_s = \cos\left(f_s(\mathbf{v}), \frac{1}{|\mathcal{Y}_s|}\sum_{y\in\mathcal{Y}_s}\mathbf{t}_y\right), \quad h_o = \cos\left(f_o(\mathbf{v}), \frac{1}{|\mathcal{Y}_o|}\sum_{y\in\mathcal{Y}_o}\mathbf{t}_y\right). \quad (2)$$

Note that we use $f_s$ and $f_o$ to decompose visual features $\mathbf{v}$, which is different from DFSP (Lu et al., 2023) that only decomposes the compositional logits. In experiments, we show the superiority of performing both visual and language decomposition in Table 5.

Following the spirit of distribution modeling, we also introduce the distributions over state and object categories, where the corresponding DSP, denoted as $\mathbf{D}^{(s)}$ and $\mathbf{D}^{(o)}$, are obtained by grouping $\mathbf{D}^{(y)}$ over $\mathcal{Y}_s$ and $\mathcal{Y}_o$, respectively. This leads to the following upper-bounded cross-entropy losses:

$$\mathcal{L}_s(x,s) = -\log\frac{\exp(h_s/\tau)}{\sum_{k=1}^{|\mathcal{S}|}\exp((h_k+h_{k,s}^{(m)})/\tau)}, \quad \mathcal{L}_o(x,o) = -\log\frac{\exp(h_o/\tau)}{\sum_{k=1}^{|\mathcal{O}|}\exp((h_k+h_{k,o}^{(m)})/\tau)}, \quad (3)$$

where $h_{k,s}^{(m)}$ and $h_{k,o}^{(m)}$ are determined the same way as $h_{k,y}^{(m)}$ in Eq. (1). See details in Appendix D.

With the individual $f_s$ and $f_o$, it is safe to have $p(y|\mathbf{v}) = p(s|\mathbf{v}) \cdot p(o|\mathbf{v})$ that induces $p(y|\mathbf{v}) \propto \exp((h_s+h_o)/\tau)$. Therefore, the recomposed logit matrix $\mathbf{H}^{(rc)} \in \mathbb{R}^{|\mathcal{S}|\times|\mathcal{O}|}$ is a Cartesian sum between $\mathbf{h}^{(s)} \in \mathbb{R}^{|\mathcal{S}|}$ and $\mathbf{h}^{(o)} \in \mathbb{R}^{|\mathcal{O}|}$, *i.e.*, $\mathbf{H}^{(rc)} = \mathbf{h}^{(s)} \oplus \mathbf{h}^{(o)\top}$, where $\mathbf{h}^{(s)}$ contains all state logits and $\mathbf{h}^{(o)}$ contains all object logits. See the red and blue squares in Fig. (4), respectively.

**Stochastic Logit Mixup**  Given the recomposed logit $h_y^{(rc)} \in \mathbf{H}^{(rc)}$ and the directly learned compositional logit $h_y$, we propose a stochastic logit mixup (**SLM**) method for decision fusion by sampling a coefficient $\lambda$ from a Beta prior distribution:

$$\tilde{h}_y = (1-\lambda)h_y + \lambda h_y^{(rc)}, \quad \lambda \sim \text{Beta}(a,b), \quad (4)$$

where $(a,b)$ are hyperparameters indicating the prior preference for each decision. In training, we replace the $h_y$ and $h_k$ of Eq. (1) with the mixed logit $\tilde{h}_y$ and $\tilde{h}_k$, respectively. In testing, we use the expectation of the Beta distribution which is $a/(a+b)$.

The insights behind the SLM are that the Beta distribution indicates a prior to $h_y$ or $h_y^{(rc)}$. It provides the flexibility of which compositional decision to trust in, and the stochasticity of the coefficient $\lambda$ inherently introduces a regularization effect in training (Carratino et al., 2022). Moreover, compared to softmax probability mixup (Huang et al., 2023), our logit mixup avoids the limitation of softmax normalization over a huge number of compositional classes, that rich information of class relationship is lost after softmax normalization according to (Bang et al., 2022). Such class relationships are even more important in the CZSL problem as indicated in (Naeem et al., 2021).

| | Method | MIT-States | | | | UT-Zappos | | | | C-GQA | | | |
|---|---|---|---|---|---|---|---|---|---|---|---|---|---|
| | | S | U | H | AUC | S | U | H | AUC | S | U | H | AUC |
| Closed | CLIP (Radford et al., 2021) | 30.2 | 46.0 | 26.1 | 11.0 | 15.8 | 49.1 | 15.6 | 5.0 | 7.5 | 25.0 | 8.6 | 1.4 |
| | CoOp (Zhou et al., 2022b) | 34.4 | 47.6 | 29.8 | 13.5 | 52.1 | 49.3 | 34.6 | 18.8 | 20.5 | 26.8 | 17.1 | 4.4 |
| | ProDA[1] (Lu et al., 2022) | 37.4 | 51.7 | 32.7 | 16.1 | 63.7 | 60.7 | 47.6 | 32.7 | – | – | – | – |
| | CSP (Nayak et al., 2023) | 46.6 | 49.9 | 36.3 | 19.4 | 64.2 | 66.2 | 46.6 | 33.0 | 28.8 | 26.8 | 20.5 | 6.2 |
| | PCVL (Xu et al., 2022) | 48.5 | 47.2 | 35.3 | 18.3 | 64.4 | 64.0 | 46.1 | 32.2 | – | – | – | – |
| | HPL (Wang et al., 2023) | 47.5 | 50.6 | 37.3 | 20.2 | 63.0 | 68.8 | 48.2 | 35.0 | 30.8 | 28.4 | 22.4 | 7.2 |
| | DFSP (Lu et al., 2023) | 46.9 | 52.0 | 37.3 | 20.6 | 66.7 | **71.7** | 47.2 | 36.0 | 38.2 | 32.0 | 27.1 | 10.5 |
| | $\mathbb{PLID}$ | **49.7** | **52.4** | **39.0** | **22.1** | **67.3** | 68.8 | **52.4** | **38.7** | **38.8** | **33.0** | **27.9** | **11.0** |
| Open | CLIP (Radford et al., 2021) | 30.1 | 14.3 | 12.8 | 3.0 | 15.7 | 20.6 | 11.2 | 2.2 | 7.5 | 4.6 | 4.0 | 0.3 |
| | CoOp (Zhou et al., 2022b) | 34.6 | 9.3 | 12.3 | 2.8 | 52.1 | 31.5 | 28.9 | 13.2 | 21.0 | 4.6 | 5.5 | 0.7 |
| | ProDA[1] (Lu et al., 2022) | 37.5 | 18.3 | 17.3 | 5.1 | 63.9 | 34.6 | 34.3 | 18.4 | – | – | – | – |
| | CSP (Nayak et al., 2023) | 46.3 | 15.7 | 17.4 | 5.7 | 64.1 | 44.1 | 38.9 | 22.7 | 28.7 | 5.2 | 6.9 | 1.2 |
| | PCVL (Xu et al., 2022) | 48.5 | 16.0 | 17.7 | 6.1 | 64.6 | 44.0 | 37.1 | 21.6 | – | – | – | – |
| | HPL (Wang et al., 2023) | 46.4 | **18.9** | 19.8 | 6.9 | 63.4 | 48.1 | 40.2 | 24.6 | 30.1 | 5.8 | 7.5 | 1.4 |
| | DFSP (Lu et al., 2023) | 47.5 | 18.5 | 19.3 | 6.8 | 66.8 | **60.0** | 44.0 | 30.3 | 38.3 | 7.2 | 10.4 | 2.4 |
| | $\mathbb{PLID}$ | **49.1** | 18.7 | **20.4** | **7.3** | **67.6** | 55.5 | **46.6** | **30.8** | **39.1** | 7.5 | 10.6 | 2.5 |

Table 1: CZSL results of Closed- and Open-World settings on three datasets. Baseline results are from published literature, where the PCVL was not evaluated on the C-GQA dataset such that we use "–" instead.

## 5 EXPERIMENTS

**Datasets and Evaluation** We perform experiments on three CZSL datasets, *i.e.*, MIT-States (Isola et al., 2015), UT-Zappos (Yu & Grauman, 2014), and C-GQA (Naeem et al., 2021), following the standard splitting protocols in CZSL literature (Purushwalkam et al., 2019; Nayak et al., 2023; Lu et al., 2023). See dataset details in the Appendix E. We report the metrics in both closed-world (**CW**) and open-world (**OW**) settings, including the best seen accuracy (**S**), the best unseen accuracy (**U**), the best harmonic mean (**H**) between the seen and unseen accuracy, and the area under the curve (**AUC**) of unseen versus seen accuracy. For OW evaluation, following the CSP (Nayak et al., 2023), we adopt the feasibility calibration by GloVe (Pennington et al., 2014) to filter out infeasible compositions.

**Implementation Details** We implement the $\mathbb{PLID}$ based on the CSP codebase in PyTorch. The CLIP architecture ViT-L/14 is used by default. Without mentioning, we generate $M = 64$ texts and augment an image with $N = 8$ views, and adopt $\text{Beta}(1, 9)$ as prior. The dropout rates of TFE and VFE are set at 0.5. We use a single NVIDIA 6000Ada GPU for training and testing. Following (Lu et al., 2023), we use Adam optimizer with base learning rate 5e-5, and steply decay it with the factor of 0.5 every 5 training epochs for a total of 20 epochs. Other details are in the Appendix E.

### 5.1 MAIN RESULTS

The results are reported in Table 1. We compare with the CZSL baselines that are developed on the same frozen CLIP model. The table shows that under both the closed-world and open-world test settings, our proposed $\mathbb{PLID}$ method achieves the best performance in most metrics on the three datasets. Note that ProDA (Lu et al., 2022) also formulates the class-wise Gaussian distributions to address the intra-class diversity, but it can only outperform CLIP and CoOp on all metrics. This indicates the importance of both diversity and informativeness for the CZSL task. On the UT-Zappos dataset, the $\mathbb{PLID}$ outperforms the DFSP in terms of S, H, and AUC by 0.6%, 5.2%, and 2.7% respectively, while inferior to the DFSP on the best unseen metric. The potential reason is that DFSP fuses the text features into the image images, which better preserves the generalizability of CLIP for the small downstream UT-Zappos dataset. Note that the HPL method uses prompt learning and recognition at both compositional and primitive levels, but it performs only slightly better than CSP and way worse than our method, indicating that traditional prompt learning helps but is not enough to adapt the CLIP model to the CZSL task.

---

[1]ProDA is re-implemented since it was originally for zero-shot learning. Limited by the GPU memory, ProDA is not applicable to the C-GQA dataset which consists of more than 278K compositional classes.

| | LID | TFE | VFE | OPT | VLPD | SLM | $H_{cw}$ | $AUC_{cw}$ | $H_{ow}$ | $AUC_{ow}$ |
|---|---|---|---|---|---|---|---|---|---|---|
| (a) | | | | | | | 35.41 | 18.56 | 17.37 | 5.56 |
| (b) | ✓ | | | | | | 37.06 | 20.43 | 18.65 | 6.50 |
| (c) | ✓ | ✓ | | | | | 37.76 | 21.07 | 19.05 | 6.62 |
| (d) | ✓ | ✓ | ✓ | | | | 37.87 | 21.09 | 19.70 | 6.95 |
| (e) | ✓ | ✓ | ✓ | ✓ | | | 38.80 | 21.67 | 19.61 | 7.01 |
| (f) | ✓ | ✓ | ✓ | ✓ | ✓ | | 38.42 | 21.69 | 20.24 | 7.31 |
| (g) | ✓ | ✓ | ✓ | ✓ | ✓ | ✓ | **38.97** | **22.12** | **20.41** | **7.34** |

Table 2: **Ablation study**. (a): the baseline that uses mean pooling of text embeddings from T5-generated sentences. (b): add distribution modeling. (c): change the mean pooling to the cross-attention. (d): augment images followed by cross-attention aggregation. (e): change T5-base LLM to the OPT-1.3B. (f): add VLPD followed by the fixed logit fusion. (g): change the fusion to a stochastic manner, which reaches to our full $\mathbb{PLID}$.

| LLM | MIT-States | | | | UT-Zappos | | | | C-GQA | | | |
|---|---|---|---|---|---|---|---|---|---|---|---|---|
| | $H_{cw}$ | $AUC_{cw}$ | $H_{ow}$ | $AUC_{ow}$ | $H_{cw}$ | $AUC_{cw}$ | $H_{ow}$ | $AUC_{ow}$ | $H_{cw}$ | $AUC_{cw}$ | $H_{ow}$ | $AUC_{ow}$ |
| T5 | 38.41 | 21.53 | **20.46** | 7.34 | **54.76** | **40.18** | 44.18 | 28.47 | 26.94 | 10.65 | 9.77 | 2.35 |
| OPT | **38.97** | **22.12** | 20.41 | **7.34** | 52.38 | 38.67 | **46.61** | **30.84** | **27.87** | **11.04** | **10.55** | **2.54** |

Table 3: Effect of LLMs on three CZSL datasets.

## 5.2 MODEL ANALYSIS

**Ablation Study** In Table 2, we show the contribution of the major components in the $\mathbb{PLID}$ model. It is clear that all components are beneficial. Here we highlight some important observations: (1) Our LID method significantly improves the performance compared to the baseline (a) and is much better than ProDA (20.43% vs 16.1% of $AUC_{cw}$) when referring to Table 1. This implies that modeling the distribution by way of ProDA is not sufficient, but language informativeness is critical and preferred for the CZSL task. (2) Rows (c)(d)(e) show that TFE, VFE, and OPT-1.3B can further achieve some performance gains. (3) Rows (f)(g) show that VLPD benefits more in the open-world setting while the SLM contributes more in the closed-world setting.

| $\mathcal{N}_s$ | $\mathcal{N}_o$ | $\mathcal{N}_y$ | $H_{cw}$ | $AUC_{cw}$ | $H_{ow}$ | $AUC_{ow}$ |
|---|---|---|---|---|---|---|
| | | | 38.44 | 21.67 | 19.53 | 6.99 |
| ✓ | ✓ | | 38.30 | 21.62 | 19.49 | 6.95 |
| | | ✓ | 38.49 | 21.90 | 19.93 | 7.20 |
| ✓ | ✓ | ✓ | **38.97** | **22.12** | **20.41** | **7.34** |

Table 4: Effect of LID on classes of states ($\mathcal{N}_s$), objects ($\mathcal{N}_o$), and compositions ($\mathcal{N}_y$).

**Effect of LLM** In Table 3, we analyze the choice of LLMs by comparing $\mathbb{PLID}$ using the pre-trained T5 (Raffel et al., 2020a) and OPT (Zhang et al., 2022a). It shows the performance varies across CZSL datasets. Note that the quality of the generated texts by OPT is much better than T5 (see examples in Appendix B), the results imply that the higher text quality on the large C-GQA dataset leads to better CZSL performance. Besides, on the UT-Zappos dataset, the better OPT does not show better closed-world performance. The reason could be that UT-Zappos is too small and its commercial shoe images do not exhibit diverse visual backgrounds.

| text | image | $H_{cw}$ | $AUC_{cw}$ | $H_{ow}$ | $AUC_{ow}$ |
|---|---|---|---|---|---|
| | | 37.94 | 20.98 | 19.67 | 6.98 |
| ✓ | | 38.40 | 21.31 | 19.99 | 7.13 |
| ✓ | ✓ | **38.97** | **22.12** | **20.41** | **7.34** |

Table 5: Effect of VLPD. The three rows indicate no decomposition, decompose text-only, and decompose both (full VLPD).

**Effect of LID** In Table 4, we further investigate at which semantic level the language-informed distribution (LID) should be applied. Denote the Gaussian distribution on state, object, and composition as $\mathcal{N}_s$, $\mathcal{N}_o$, and $\mathcal{N}_y$, respectively. The Table 4 results clearly show the superiority of applying LID on all three semantic levels. This indicates the generality of language-informed distribution towards many potential zero-shot or open-vocabulary recognition problems.

**Design Choice of VLPD** In Table 5, we validate the design choices of VLPD, including the model without primitive decomposition, only decompose text into primitives, and our decomposition on both visual and language primitives (VLPD). The results show the clear advantage of our VLPD design choice. Note that DFSP also has primitive decomposition but only on text modality. Our better performance thus indicates the need for decomposition on both visual and image.

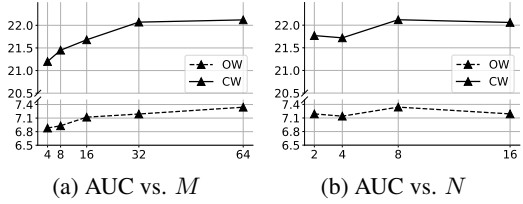
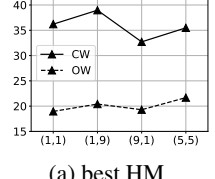
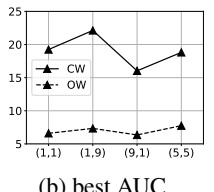

(a) AUC vs. $M$     (b) AUC vs. $N$     (a) best HM     (b) best AUC

Figure 5: Impact of $M$ and $N$. We set $N = 8$ for the Fig. 5a, while we set $M = 64$ for the Fig. 5b.

Figure 6: Impact of $(a, b)$. Here $(1, 1)$ implies random sampling while $(5, 5)$ implies equally trusted.

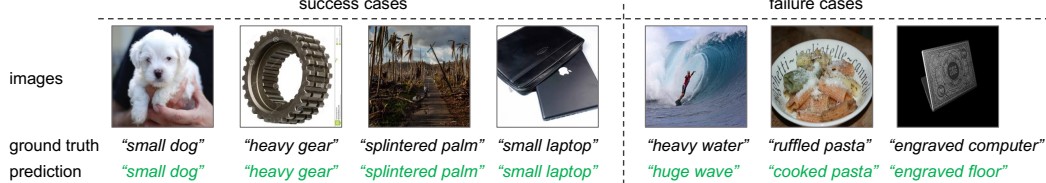

| | success cases | | | | failure cases | | |
|---|---|---|---|---|---|---|---|
| images | | | | | | | |
| ground truth | *"small dog"* | *"heavy gear"* | *"splintered palm"* | *"small laptop"* | *"heavy water"* | *"ruffled pasta"* | *"engraved computer"* |
| prediction | *"small dog"* | *"heavy gear"* | *"splintered palm"* | *"small laptop"* | *"huge wave"* | *"cooked pasta"* | *"engraved floor"* |

Figure 8: Qualitative results. We show the success and failure cases of prediction on the MIT-States test set.

**Hyperparameters** In Fig. 5, we quantitatively show the impact of the number of generated text descriptions $M$ and the number of augmented image views $N$. It shows that the best performance is achieved when $M = 64$ and $N = 8$. We note that more augmented image views slightly decrease the performance, which could be attributed to the overfitting of the seen compositions.

In Fig. 6, we show the impact of the Beta prior parameters $(a, b)$. We set them to $(1, 1)$ for random sampling, $(1, 9)$ for preference to the composition, $(9, 1)$ for preference to re-composition, and $(5, 5)$ for equal preference, respectively. It reveals that trusting more of the directly learned composition by $\text{Beta}(1, 9)$ achieves the best results.

**Qualitative Analysis** We use the tSNE to visualize the generated text embeddings $\mathbf{D}$ and the learned DSP from or $\mathbb{PLID}$ model in Fig. 7, where the same set of 10 compositional classes are randomly selected from MIT-States dataset. It shows that by learning the distribution of each composition from LLM-generated texts using Eq. (1) and (3) and TFE module, compositional class embeddings can be distributed more compactly in each class (small intra-class variance),

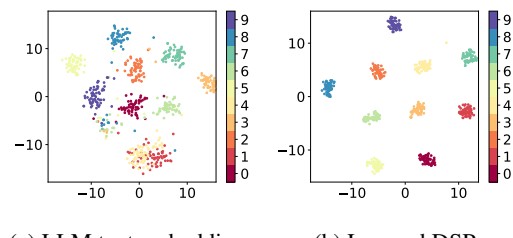

(a) LLM text embeddings     (b) Learned DSP

Figure 7: tSNE visualization of the text embeddings.

and better separated among multiple classes (large inter-class distance). In Appendix F, we show primitive-level tSNE embedding visualizations that reveal the same observation.

In Fig. 8, we show some success and failure cases of our $\mathbb{PLID}$ model. For example, the `heavy water` case indicates an incorrect label while $\mathbb{PLID}$ could correctly predict it as `huge wave`. This shows the robustness of $\mathbb{PLID}$ against noisy labels. The last two failure cases reveal $\mathbb{PLID}$ still could make mistakes on the state prediction (`cooked` `pasta`) and object prediction (`engraved` `floor`), which indicates there is still a long way to go for the CZSL problem.

## 6 CONCLUSION

In this work, we propose a novel CLIP-based compositional zero-shot learning (CZSL) method named $\mathbb{PLID}$. It leverages the generated text description of each class from large language models to formulate the class-specific Gaussian distributions. By softly prompting these language-informed distributions, $\mathbb{PLID}$ could achieve diversified and informative class embeddings for fine-grained compositional classes. Besides, we decompose the visual embeddings of image data into simple primitives that contain the basic states and objects, from which the re-composed predictions are derived to calibrate the prediction by our proposed stochastic logit mixup strategy. Experimental results show the superiority of the $\mathbb{PLID}$ method to prior arts on all common CZSL datasets.

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

## A   BROADER IMPACT AND LIMITATIONS

**Broader Impact**   This work can be broadly extended to more downstream multi-modality applications, such as general zero-shot learning, text-image retrieval, text-to-image generation, etc., when the class composition is not especially taken into consideration. Besides, the central idea of LLM-grounded modality alignment is not limited to text and image, but any modality that could reveal the semantic categories in practice is promising to explore in the future. The potential negative societal impact is that, the developers should be cautious by carefully examining the societal biases indicated by the generated textual class descriptions, even though the large language models we used are publicly accessible.

**Limitations**   One limitation is that the primitive decomposition could be difficult to learn when the states are non-visual concepts like `smelly`, `hot`, etc., even by the pre-trained CLIP model. Another limitation is that the generated descriptions by LLMs are not grounded to the image such that some distraction from generated descriptions could be introduced.

## B   GENERATING COMPOSITIONAL CLASS DESCRIPTIONS

In this work, we choose T5 and OPT models as the LLMs for compositional class description generation. For the T5 model, we follow the same setting as (He et al., 2023) that uses the T5-base model for word-to-sentence generation. The T5-base model was pre-trained on the Colossal Clean Crawled Corpus dataset (Raffel et al., 2020b) and finetuned on the CommonGen dataset (Lin et al., 2020). Take the `painted ceiling` as an example, the results from T5-base model are:

```
– A very old but beautifully decorated ceiling.
– A remodeled interior with a painted ceiling.
– A painted ceiling at a restaurant.
– Stained glass windows and a carved pattern on the ceiling.
– Painted ceilings and a fireplace.
– This apartment has a painted ceiling.
– A painted ceiling was a huge hit.
– A chandelier is painted in the middle of the ceiling.
– A stained glass window in a bathroom with a white painted ceiling.
– The ceiling of a hotel is painted.
– ...
```

For the OPT model, we adopt the pre-trained OPT-1.3B model which is small but still could generate sentences of sufficiently good quality. Considering that it is a decoder-only LLM that relies on a few examples as the prompt for word-to-sentence generation, we design the prompt template with 4 examples as shown below.

```
Keywords: sliced, potato, picture
Output: The picture features a beautifully arranged plate of thinly
    sliced potatoes.
###
Keywords: red, apple, photo
Output: In the photo, a bright red apple is the central focus,
    captured in stunning detail.
###
Keywords: leather, shoes, image
Output: The image showcases a sleek pair of leather shoes,
    meticulously designed to impress.
###
Keywords: broken, car, photo
Output: The photo captures the aftermath of a car accident with a
    broken vehicle lying on the road.
###
```

where the `Keywords` is followed by the words of the state, object, and the word randomly picked from the set {`photo`, `image`, `picture`}. Using the same example *painted ceiling* as T5-base model, the generated sentences are:

- The painting of the ceiling features an intricate pattern of
    intricate gold-colored paint and is framed by a white
    background.
- The ceiling has been painted with the pictures of these three
    characters, all arranged together.
- In the picture, the ceiling is covered in bright, colorful
    paintwork that has been painted on by hands painted white. The
    colors have been selected carefully.
- In the picture, the ceiling features painted decoration. The
    decoration resembles the surface of the sea, and has been
    painted in shades of blue.
- The photograph captures both the bright colors of the painting
    atop the ceiling and the subtle shades of light reflecting off
    of it.
- The large picture shows a large pattern painted onto the ceiling.
    The blue line shows paint dripping down.
- The wall behind the picture shows three different painted
    ceilings, in bright contrasting colors. A vibrant sky and blue
    skies are depicted against the dark brick wall.
- The ceiling of the room depicted in the painting could very well
    be painted in a few hours. The details of each object are
    clearly defined in its placement and position.
- Another photo of the same scene, this time featuring a ceiling
    painted in a stunning, white color.
- A painted ceiling is shown, painted according to a specific
    design. this is a typical design that can also include
    decorative or functional elements.
- ...

It is clear that the generated class descriptions are much more diverse and informative than those of the OPT model.

## C  COVARIANCE SHARING

For the CZSL task, the spatial complexity of computing the covariance matrix $\boldsymbol{\Sigma}_{1:C}$ is $O(|C^{(s)}|^2 d)$ which could be too heavy to compute if the number of the compositions is too large. For example, the C-GQA dataset contains 278K seen compositions which result in around $6 \times 10^{13}$ floating elements of $\boldsymbol{\Sigma}_{1:C}$ for 768-dim text features. To handle this issue, we instead implement the $\boldsymbol{\Sigma}_{1:C}$ by sharing the covariance across attributes given the same object. This implies that the model is encouraged to learn the object-level distributions.

Specifically, similar to the VLPD module of the main paper, we compute the mean $\boldsymbol{\mu}_{1:|\mathcal{O}|}$ and covariance $\boldsymbol{\Sigma}_{1:|\mathcal{O}|}$ over the objects by grouping $\mathbf{t}_y$ and $\mathbf{D}^{(y)}$ with object labels:

$$\mathbf{t}_o = \frac{1}{|\mathcal{Y}_o|} \sum_{y \in \mathcal{Y}_o} \mathbf{t}_y, \quad \mathbf{D}^{(o)} = \frac{1}{|\mathcal{Y}_o|} \sum_{y \in \mathcal{Y}_o} \mathbf{D}^{(y)}, \tag{5}$$

where $\mathcal{Y}_o$ is the subset of compositions in $\mathcal{Y}$ that contains the same object as $y$. Then, all the pairwise margins $\mathbf{H}_o^{(m)} \in \mathbb{R}^{|\mathcal{O}| \times |\mathcal{O}|}$ in object space can be mapped back to $\mathbf{H}^{(m)} \in \mathbb{R}^{C \times C}$ in a compositional space by sharing it with all compositions in $\mathcal{Y}_o$. This could significantly reduce the computation load of the covariance while compromising the accuracy of distribution modeling.

Since the distribution modeling for both our $\mathbb{PLID}$ and ProDA is not applicable to the C-GQA dataset, we use the MIT States dataset to show the negative impact of sharing the covariance (see Table 6). It shows that the covariance sharing can significantly save the GPU memory (17.6 vs 32.5 GB), while still performing much better than ProDA.

## D  PRIMITIVE-LEVEL GAUSSIAN MODELING

To formulate the Gaussian distributions over the state classes and the object classes, we group the text embeddings of composition descriptions $\mathbf{D}$ by Eq. (5), resulting in the distribution support points

| Variants | Mem.(GB) | $H_{cw}$ | $AUC_{cw}$ | $H_{ow}$ | $AUC_{ow}$ |
|---|---|---|---|---|---|
| ProDA (Lu et al., 2022) | 32.5 | 32.71 | 16.11 | 17.30 | 5.11 |
| PLID (w. ShareCov) | **17.6** | 38.50 (-0.47%) | 21.69 (-0.43%) | 19.81 (-0.60%) | 7.04 (-0.30%) |
| $\mathbb{PLID}$ (full) | 22.2 | **38.97** | **22.12** | **20.41** | **7.34** |

Table 6: Effect of covariance sharing on MIT-States dataset. All methods use the same batch size of 64 for a fair comparison of GPU memory.

(DSP) $\mathbf{t}_o + \mathbf{D}^{(o)}$ and $\mathbf{t}_s + \mathbf{D}^{(s)}$ for a given object class $o$ and state class $s$, respectively. The DSPs are assumed to follow the state distribution $\mathcal{N}(\mathbf{t}_s, \boldsymbol{\Sigma}_s)$ or the object distribution $\mathcal{N}(\mathbf{t}_o, \boldsymbol{\Sigma}_o)$, where the covariances $\boldsymbol{\Sigma}_s$ and $\boldsymbol{\Sigma}_o$ are determined by $\mathbf{D}^{(s)}$ and $\mathbf{D}^{(o)}$, respectively.

Eventually, given the decomposed state visual features $f_s(\mathbf{v})$ and object visual features $f_o(\mathbf{v})$, the logit margin terms are defined as

$$h_{k,s}^{(m)} = f_s(\mathbf{v})^\top \mathbf{A}_{k,s} f_s(\mathbf{v}), \quad \text{and} \quad h_{k,o}^{(m)} = f_o(\mathbf{v})^\top \mathbf{A}_{k,o} f_o(\mathbf{v}), \tag{6}$$

where the index $k$ ranges within $[1, |\mathcal{S}|]$ for computing the state classification loss $\mathcal{L}_s$, and ranges within $[1, |\mathcal{O}|]$ for computing the object classification loss $\mathcal{L}_o$, respectively.

# E   MORE IMPLEMENTATION DETAILS

**Datasets**   We perform experiments on three CZSL datasets, *i.e.*, MIT-States (Isola et al., 2015), UT-Zappos (Yu & Grauman, 2014), and C-GQA (Naeem et al., 2021). MIT-States consists of 115 states and 245 objects, with 53,753 images in total. Following (Purushwalkam et al., 2019; Nayak et al., 2023; Lu et al., 2023), it is split into 1,262 seen and 300/400 unseen compositions for training and validation/testing, respectively. UT-Zappos contains 16 states and 12 objects for 50,025 images in total, and it is split into 83 seen and 15/18 unseen compositions for training and validation/testing. C-GQA contains 453 states and 870 objects for 39,298 images, and it is split into 5,592 seen and 1,040/923 unseen compositions for training and validation/testing, respectively, resulting in 7,555 and 278,362 target compositions in closed- and open-world settings.

**Implementation**   Our model is implemented on top of the CSP (Nayak et al., 2023) codebase, which extends the CLIP model for compositional zero-shot learning. To tokenize the generated long sentences of each compositional class, we set the context length to the default value of 77 in the original CLIP model. For the soft prompt embeddings, we set the context length of text encoder to 8 for all datasets. We use the dropout rate of 0.3 for the learnable state and object embeddings. In training, we follow the DFSP (Lu et al., 2023) that uses the performance of the validation set for model selection. The rest hyperparameters of our final model on each dataset are listed in Table 7.

| Hyperparameters | MiT-States | UT-Zappos | C-GQA |
|---|---|---|---|
| max epochs | 50 | 25 | 20 |
| base learning rate | 0.00005 | 0.0001 | 0.00001 |
| weight decay | 0.00002 | 0.00001 | 0.00001 |
| number of text descriptions | 64 | 32 | 64 |
| number of image views | 8 | 8 | 8 |
| attention dropout | 0.5 | 0.1 | 0.1 |
| weights of primitive loss | 0.1 | 0.01 | 0.01 |

Table 7: Hyperparameters of model implementation.

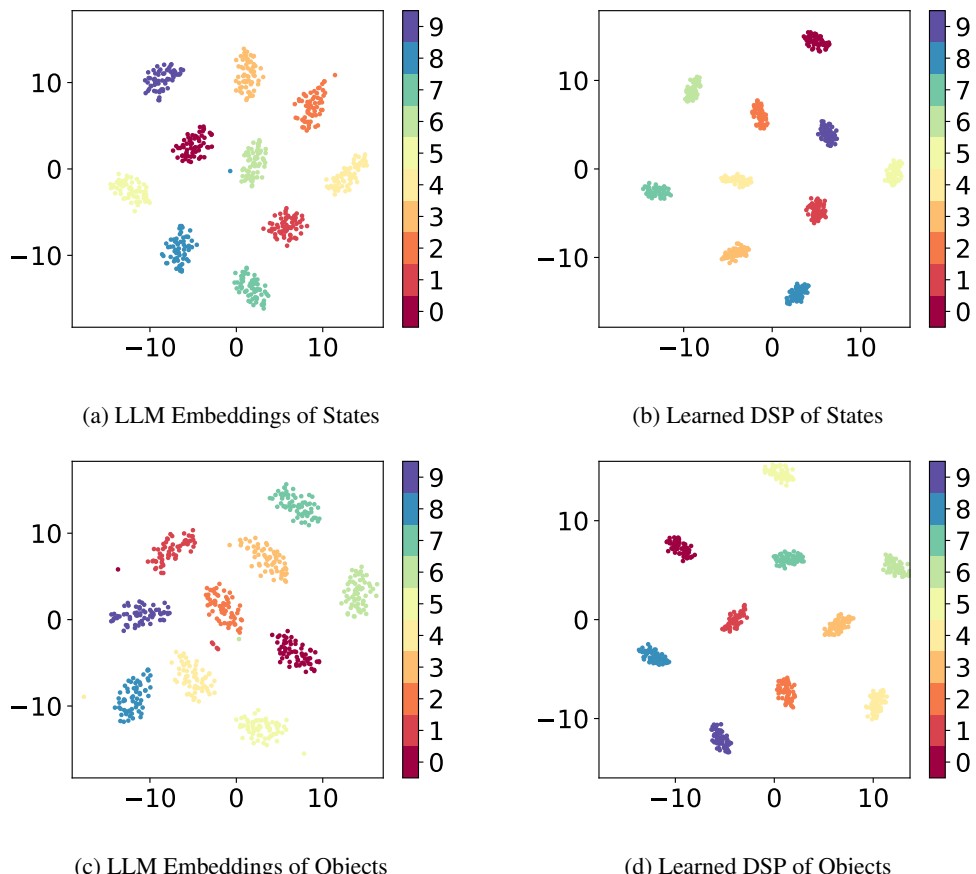

(a) LLM Embeddings of States

(b) Learned DSP of States

(c) LLM Embeddings of Objects

(d) Learned DSP of Objects

Figure 9: tSNE visualization of the primitive-level text embeddings (*states*: Fig. 9a and 9b, *objects*: Fig. 9c and 9d). This figure clearly shows that, compared to the raw embeddings by pre-trained LLMs, our method achieves better distributions over both the state and object classes.

## F   MORE RESULTS

**Primitive-level Visualization**   In addition to the tSNE visualization of Gaussian distributions over the composition-level classes, we provide the visualizations of the primitive-level classes in Fig. 9. These figures show that our model could learn better text distributions over state classes and object classes than those of the pre-trained LLMs.

