# OpenReview forum: "Prompting Language-Informed Distribution for Compositional Zero-Shot Learning"
_ICLR.cc/2024/Conference — Submitted to ICLR 2024_

### Official Review · Reviewer_feoJ · 2023-10-30

**Soundness:** 3 good
**Presentation:** 3 good
**Contribution:** 3 good
**Rating:** 5
**Confidence:** 5

**Summary:**

In this work, the authors utilize pre-trained large language models (LLM) to increase the diversity of the semantic descriptions so that the the visual and semantic features can be aligned accurately. Moreover, a stochastic logit mixup (SLM) strategy are proposed for the final compositional predictions. The experiments demonstrate that the proposed methord improves the current benchmark performance on three benchmark datasets.

**Strengths:**

1.The paper is organized and clearly written.
2.The proposed method seems to be intuitively reasonable.

**Weaknesses:**

1.The proposed method relies much on the quality of LLMs, and the transferability of the model is not reflected in the paper.
2.According to the Ablation study, the experiment w/o VLPD does not change much (even the H_cw value decreases).
3.The proposed languageinformed distributions (LID) can effectively avoid the issue of intra-class variety. However, the authors would better also intepret how to solve the issue of inter-class correlation.

**Questions:**

NA

---

> ### Author Response · Authors · 2023-11-21
> **Authors' response to Reviewer feoJ**
>
> Thanks for reviewing the paper. Below we present our responses to the raised concerns.
>
> -  **Rely on the LLM quality.** We would clarify that leveraging LLMs for the CZSL task is one of our proposed ideas in this paper, and it has never been explored in the CZSL literature. Therefore, whether the CZSL performance replies on LLM quality or not is an open research question, and interestingly, we found the performance is not always better if a better LLM is utilized (see Table 3). We believe in the future, there will be more research investigating how to better utilize LLMs to solve the challenging CZSL task.
>  -  **Impact of VLPD.** The VLPD primarily improves the open-world CZSL performance ($H_{ow}$ and $\text{AUC}_{ow}$), rather than the closed-world ones. This can be observed in Table 2 which indicates significant open-world performance gains. This observation is expected because the VLPD is developed to mitigate the overfitting issue on the visual data of seen compositions. Thus, in a much larger open-world decision space where most compositional class names might be infeasible, a less overfitting model w.r.t. the seen compositions is expected to achieve higher open-world performance.
>  - **Inter-class correlation.** We clarify that the proposed LID is simultaneously minimizing the intra-class variance and maximizing the inter-class separability as stated in the Lines below Eq. (1) (the last 3 lines of Page 5). More specifically, the learning goal is to minimize the NLL upper-bound (r.h.s of Eq. (1)). This will encourage minimizing the $h_{k,y}$ term which is determined by the positive intra-class variance terms $\Sigma_{kk}$ and $\Sigma_{yy}$, and also the negative inter-class terms $-\Sigma_{ky}$ and $-\Sigma_{yk}$. It's thus clear that inter-class correlation is also optimized.

---

### Official Review · Reviewer_4PfY · 2023-10-31

**Soundness:** 3 good
**Presentation:** 3 good
**Contribution:** 3 good
**Rating:** 6
**Confidence:** 3

**Summary:**

This paper proposes a CLIP-based compositional zero-shot learning (CZSL) method. Two components are introduced i.e., primitive decomposition and stochastic logit mixup to fuse the classification decision from compositional and primitive predictions. Finally, the suggested method shows the performance improvement of the PLID method to prior arts on all common CZSL datasets.

**Strengths:**

The motivations of suggested primitive decomposition and stochastic logit mixup are okay, which have positive effects on zero-shot compositional visual recognition.

The paper is well presented and the suggested method slightly exceeds the comparison method.

**Weaknesses:**

The authors did not mention whether the comparison methods also used N-view augmentation. If the answer is no, I think the comparison is unfair. Please indicate which method uses the same augmentation.

The author mentioned in the ablation study that LID significantly improves performance compared to the baseline is confusing. According to Table 2, TFE and VFE seem to have little effect, while LID appears to be effective, and its effectiveness should come from LLM, which in this paper belongs to incremental engineering.

Figure 2 should be consistent with its caption. For example, Figure 2 should provide the corresponding parts of LID, which should be more helpful for understanding.

**Questions:**

Please refer to the Weaknesses.

---

> ### Author Response · Authors · 2023-11-22
> **Authors' response to the reviewer 4PfY**
>
> We sincerely thank the reviewer's acceptance of this paper and all the efforts in reviewing it.
>
> 1. **$N$-view augmentation**
>
> We would clarify that the comparison is fair though it is not used in comparison methods. The reason is that the N-view augmentations on the images correspond to the motivations of our distributional modeling, that introducing variety/uncertainty into visual-text alignment by TFE and VFE leads to better zero-shot generalization. This is the contribution of our method that the VFE depends on, rather than a general technique that can be adopted by other methods.
>
> To further validate its effectiveness, we compare the model without $N$-view augmentation with the SoTA method DFSP as shown below. It shows that our method still performs better than (or comparable with) DFSP.
>
> | models | $\text{H}_{cw}$ | $\text{AUC}_{cw}$ | $\text{H}_{ow}$ | $\text{AUC}_{ow}$ |
> | :---  | :----: | ----: | ----: | ----: |
> | DFSP | 37.3 | 20.6 | 19.3 | 6.8 |
> | Ours (w/o N-views) | 37.89 | 21.07 | 19.37 | 6.78 |
>
> 2. **Performance of LID**
>
> Note that the LLM is the core component to form the LID. Besides, using LLM-generated text as class distribution points in prompt learning is a new paradigm for zero-shot learning, which has never been explored in literature, rather than incremental engineering. The performances of LID and LLM are individually analyzed in the response to the Reviewer BPd3 (see the Table in response), which shows the contribution from both especially in the closed-world CZSL setting.
>
> 3. **Figure 2**
>
> Note that the figure has provided the LID part (top-center part). We will revise it to make the LID more noticeable for readers.
>
> Thank you again for your constructive comments!

---

### Official Review · Reviewer_BPd3 · 2023-11-01

**Soundness:** 3 good
**Presentation:** 2 fair
**Contribution:** 2 fair
**Rating:** 5
**Confidence:** 4

**Summary:**

The article proposes an approach for compositional zero-shot learning (CZSL) where the goal is to recognize compositions of attributes and objects, having only a subset of them during training. To address this task, the article proposes to exploit various components:
1. Language-informed distribution (LID) which exploits composition-specific prompts given by an LLM and modifies them via learned soft-prompts. Both soft and LLM-based prompts are fed to a cross-attention module (TFE), and the resulting vector is encoded as a distribution, where the vector is the mean and the same with added LLM-based embeddings define other data points.
2. On the visual side, visual embeddings are obtained by augmenting the input via multiple views and passing them (and the original input) through a cross-attention module (VFE). Both VFE, TFE, and soft-prompts are learned via cross-entropy loss over the compositional labels and taking into account a margin to encourage inter-class separability.
3. To decompose the compositional elements, the article introduces VLPD which computes classification loss over objects and attributes independently, where each primitive logit is derived by marginalizing compositional embeddings.
4. Finally, Stochastic Logic Mixup is proposed to mix predictions coming from primitive-specific and composition-specific logits.

Experiments over various datasets (e.g. MIT-states, C-GQA, UT-Zappos) show the superiority of the approach w.r.t. the previous state of the art, with ablation studies confirming the efficacy of the proposed modules.

--------

**Update post-rebuttal:**

I thank the authors for their response. At the same time, the additional analyses partially addressed the initial concerns, e.g. removing N-views shows results comparable to other models (and vice-versa, how models would improve from n-views it is not tested), there is a limited performance gap between different variants with/without the contribution, and unclear how writing issues will be improved. I value LID per-se as a contribution but, for the reasons outlined above, I deem the article to be borderline for this venue and I tend to keep my initial score.

**Strengths:**

1. Despite the method having several building blocks, the idea and motivation behind each introduced component are well-described in the text: e.g. LLM is used to create a pool of sentences describing the compositions, which in turn is used to estimate a distribution that can be in turn used to estimate pairwise-margins between compositions. Fig. 1 also helps the reader understand the starting idea for the model (i.e. composition ambiguities), and why a distribution over the language space may help deal with such uncertainty.
2. The experiments show that the proposed approach (PLID) surpasses by a margin all competitors in all settings, especially in the closed-world scenario where unseen compositions are known (Table 1). In this latter setting, the gap in AUC is remarkable, with 1.5 points improvement on MIT-states, 2.7 on UT-Zappos, and 0.5 on C-GQA.
3. Sections 2, 3, and 4 give credit to the approaches the method builds on, presenting a detailed overview of the literature and the proposed contributions.

**Weaknesses:**

I have two main concerns regarding the experimental analysis and the presentation.

For the former, the proposed method contains several components (i.e. LID, TFE, VFE, VLPD, SLM). While all components have motivations justifying their use, each of them has specific design choices whose impact is not fully clear from Section 5.2. Examples are:
1. The gap between considering and not considering distributional-based margins in Eq. (1) and Eq. (4) is mild accordingly is mild according to Table 4 (i.e. gap lower than 0.5 points but for harmonic mean OW). Fig. 5.a shows that indeed going from 4 to 64 LLM-based sentences improves the overall results but less than 1 point and less in the more challenging OW setting. Given that querying LLM is costly in this setting (i.e. the number C of compositions might be in the order of thousands) and can be noisy (as per Appendix A), it is questionable whether LLMs and distributional semantics are crucial for the approach (as suggested by the title). The ablations are also conducted on the MIT-states dataset which is known to be noisy (Atzmon et al. 2020), thus it should be verified if the findings hold across datasets.
2. Related to the previous points, TFE and VFE are modules that refine visual/textual embeddings. The article does not contain ablations on their number of parameters/design choices and the improvement from the added views is mild (i.e. less than 0.5 points on Fig. 5b). Ablating variants of these modules, potentially taking out the set of text embeddings/views and focusing on their parameters (e.g. even via MLPs, etc.) would strengthen the need for their implementation as cross-modal blocks and also of their specific input choices. This applies also to the specific implementations of the prediction modules $f_s$ and $f_o$ in Eq. (2).
3. The SLM module should provide flexibility to the model regarding which predictions to trust. However, SLM is not compared to other aggregation strategies (e.g. average, max, product) and on the single scoring mechanism (compositional vs primitive-based). Thus, it is hard to assess the need for this module.
4. Regarding the presentation: from the abstract and introduction it is unclear what is the role of LLMs/how they are used. It is implied that a language-informed class distribution is used, but not how this is achieved (or kept this information generic in Section 1). This is not a major weakness per-se, but given that the title focuses on this distribution, it would be helpful to give hints on how this distribution is estimated already at the beginning of the manuscript, clarifying the methodological idea to the reader. This is a purpose that Fig. 1 serves well, but the text does not stress.
5. The notation of Section 3 is not straightforward to follow. The main reason is not the lack of explanations (each term is properly defined) but the number of terms defined that the reader should remember to fully appreciate the method. While I understand that it is not easy to make the notation simpler given the presence of multiple components, in some cases, the notation could be simplified. For instance, the name of the text embeddings is detached from their inputs (e.g. S becomes D, [p:..] becomes q, x and X become v, etc.). The end of the VLPD part introduces $\mathbf{h}$ elements and $\mathbf{H}$, that could be replaced by simply stating that $h^{rc}_y = h_s + h_o$ (even directly on Eq. (4)). These are (arguable and probably subjective) examples on how some of the elements could be not defined and/or the notation simplified.

Minors:
- The qualitative results in Fig. 8 do not provide specific insights on the model as the predictions are only compared with the ground truth. As in Fig. 7, it would have been more helpful to investigate how predictions are affected by different design choices.
- Fig. 2, caption, "VEF" vs "VFE".

**Questions:**

Overall, I like the principles and ideas behind the approach. At the same time, the concerns regarding the experimental validation of the design choices should be answered in the rebuttal. In particular:
1. What is the impact of the LLM and the cost to produce the compositional descriptions?
2. Do the structure of TFE and VFE matter more or less than the distribution/margins introduced?
3. Is SLM better than simpler aggregation strategies?

**Details Of Ethics Concerns:**

None.

---

> ### Author Response · Authors · 2023-11-22
> **Authors' response to Reviewer BPd3**
>
> We sincerely thank the reviewer's support for the principles and ideas behind the approach. The comments are constructive to improve the quality of this work.
>
> **1. Experimental validation of design choices.**
>
> We would like to emphasize that the design choices of TFE, VFE, $f_s$, and $f_o$ are not the contribution of this paper. To address the concerns about their impact, here we provide the results of the model with different design choices in the Table below. Specifically, for simplicity, we keep the structures of TFE and VFE the same and replace their default one-layer cross-attention (XAttn$\times 1$) implementations with three-layer cross-attention (XAttn$\times 3$) or one-layer MLP (MLP$\times 1$). For the SLM module, we replace it with the version that only uses compositional logits (comp.-only logit), and the version with simple weighted averaging (w-avg. logit) by setting $\lambda$ to 0.1 (the same as the expectation of the Beta distribution in our SLM).
>
> | design choices      | $\text{H}_\text{cw}$ | $\text{AUC}_\text{cw}$  | $\text{H}_\text{ow}$ | $\text{AUC}_\text{ow}$     |
> | :---        |    :----:   |  :----:   |   :----:   |   :----:   |
> | w/o LLM, w/o LID  | 38.01  | 21.00  | 19.64  |  6.93 |
> | LLM, w/o LID | 38.44  | 21.67  | 19.53  | 6.99 |
> | MLP ($\times 1$) | 38.99 | 22.02 | 20.45  | 7.37 |
> | XAttn ($\times 3$) | 37.46  | 20.65  | 19.15  | 6.70  |
> | comp.-only logit  | 37.94     | 20.98 | 19.67 |  6.98 |
> | w-avg. logit ($\lambda=0.1$)  | 38.67 | 21.90 | 19.99 | 7.15 |
> | **PLID** (LLM, LID, XAttn$\times 1$, SLM)  | 38.97 | 22.12  | 20.41 | 7.34 |
>
> We have the following observations.
>
>  - **LLM indeed has a noticeably positive effect while the cost is acceptable.** For the CZSL evaluation especially the very challenging open-world metrics, it is normal that the scales of numbers are very low and even 0.5 points are meaningful in practice (see other methods in Table 1). If removing LLM descriptions (w/o LLM, w/o LID), we present results in Table above (row (1)) which shows the closed-world performance decreases significantly ($-1.12$ points of $AUC_{cw}$). With LLM added but still without LID, the closed-world performance further improved ($+0.67$ points of $AUC_{cw}$). These results demonstrate that LLM and LID could collectively improve the CZSL performance. Regarding the concern on the cost of LLMs, in practice, the compositional descriptions are generated offline by frozen LLM and their features are pre-extracted by frozen CLIP so that the cost for our model training and testing is acceptable.
>  - **Both the structure of TFE/VFE and the introduced distribution matter.** Comparing the result row (2)(3)(4) with our full model (the last row), it shows that without the distribution modeling by LID (LLM, w/o. LID), the results (row (2)) are worse than models with LID but instead using simple MLP (row (3)), while better than the model with more cross-attention (XAttn$\times 3$) layers (row (4)). This indicates that the number of TFE/VFE layers matters more but the types (attn/mlp) matter less than the LID. Since LID has never been explored in CZSL literature, these findings could inspire future exploration on how to better utilize LLM-based distributional semantics.
>  - **SLM is better than simpler (or without) logit aggregation.** In row (5), we simply use compositional logits without primitive ones, while in row (6), we use simple weighted averaging. We see that neither of these simple methods could achieve better performance and noticeably, only using compositional logits lags far behind our SLM strategy which indicates the importance of primitive-based logits in the CZSL task.
>
> **2. Presentation and Notations**
>
> Thanks for providing the valuable writing suggestions! We will revise the Abstract and Introduction text part to convey clearly the methodological idea. For the notations, we have intended to precisely convey the technical truth of our method through the notations, while not being aware of their overloaded information for readers. Thanks to your suggestion, we will try our best to simplify them. We will also revise the figures in the final version.

---

> ### Comment · Reviewer_BPd3 · 2023-11-22
> **Thank you and further clarifications**
>
> I thank the authors for their response. I would like to ask for further clarifications:
>
> 1. From the table, the MLP (x1) row achieves better results than the full model in 3 out of 4 metrics. Is it using SLM and VLPD? and what happens if the single MLP is replaced by the single cross-attention layer (without SLM and VLPD)? I am asking this question because the contribution (b) at the end of the introduction depends on the impact of these two components.
> 2. The ablation is reported on MIT-states but a) the results of all models are very close and b) this dataset is noisy (Atzmon et al. 2020). As I mentioned in point 1, it would have been helpful/made the contribution stronger to verify if the improvement brought by each component holds across datasets as well. Is there a reason for performing the ablation only on MIT-states?
> 3. While I appreciate the will to update the manuscript to improve its clarity, it would have been better to update the article as well (given that ICLR allows the upload of revisions). Is this due to time constraints and/or some changes that are hard to incorporate?
> 4. It would also be helpful answering to the concerns of Reviewer 4PfY, pointing to potential mismatches in the number of views in the comparisons.

---

> ### Author Response · Authors · 2023-11-22
> **Response to the further questions.**
>
> 1. The MLP (x1) row keeps all others the same (uses SLM and VLPD) except for the TVE/VFE module, where we replace the cross-attention with MLP. Though it performs better, the improvement is rather small (less than 0.1 points). This is why we conclude that the type of the structure does not matter. For such structural exploration on the model without SLM and VLPD, we can do that but the inner structure of the module is NOT our claimed contribution and it would take more time at this moment. For your concern about our claimed contribution (b), we already have the variable-controlled experimental results in Table 2 (row (f)(g)) that validate the VLPD and SLM.
>
> 2. In terms of the ablation study on more datasets, we are currently doing the ablation study on the UT-Zappos dataset. At this moment, we already have the new results by individually removing LLM, LID, VLPD, and SLM which are the major claimed technical contributions.
>
> | Ablations | $\text{H}_{cw}$ | $\text{AUC}_{cw}$ | $\text{H}_{ow}$ | $\text{AUC}_{ow}$ |
> | :---  | :----: | :----: | :----: | :----: |
> | w/o LLM | 43.47 | 30.11  | 38.44  | 23.16 |
> | w/o LID | 50.27 | 35.97 | 43.50 | 26.94 |
> | w/o VLPD | 53.58 | 36.61 | 42.59 | 25.76 |
> | w/o SLM | 49.77 | 36.24 | 39.55 | 24.93 |
> |Full Model| 52.38 | 38.67 | 46.61 | 30.84 |
>
> This table shows that the proposed components indeed work well on the UT-Zappos dataset. We will add this Table to the main paper.
>
> 3. Yes, we appreciate your understanding of the time constraints and workload needed to improve the clarity. Actually, we just finished the CVPR'24 submission work and put all my effort into the ICLR rebuttal and experiments these days. We promise that the substantial revision to improve the clarity will be done later.
>
> 4. For the reviewer 4PfY's question, we clarify that there is no such mismatch problem. The reason is that the N-view augmentations on the images correspond to the motivations of our distributional modeling, that introducing variety/uncertainty into visual-text alignment by TFE and VFE leads to better zero-shot generalization. This is the contribution of our method that the VFE depends on, rather than a general technique that can be adopted by other methods.

---

### Official Review · Reviewer_FS69 · 2023-11-02

**Soundness:** 2 fair
**Presentation:** 2 fair
**Contribution:** 2 fair
**Rating:** 5
**Confidence:** 5

**Summary:**

The paper proposes prompting the language-informed distribution, i.e., PLID, for handling the CZSL task. PLID presents language-informed class distributions that are diverse and informative and enhance the compositionality of the class embedding. Visual-language primitive decomposition and stochastic logit mixup strategy are used to fuse the decisions of the compositional and the primitive logit space. Experimental results demonstrate that the PLID method effectively improves the performance.

**Strengths:**

The overall organization is reasonable, and the writing is good.

The class-wise distribution modeling and its afterward alignment with the image modal is novel.

Sufficient experiments and ablation studies are performed.

**Weaknesses:**

The motivation for modeling class distribution is not new, i.e., it has been proposed and used in the vision-language model. Also, the augmentation to image input seems like a test-time adaptation strategy that is also explored in the community. These aspects degrade the contribution to the community.

The framework incrementally follows ProDA by introducing D^{(y)} and introduces VLPD by compositing v which is also widely used in CZSL. The framework seems like a combination of many existing techniques. This degrades its novelty. Why A in Eq. 1 is defined like that? Is the dimension shape of A correct by defining it as A_k,y? What’s the shape of A_k,y?

Some parameter analysis is shown, however, the results w.r.t. the value of N = 0 is not shown. Also, how to balance the tradeoff between different losses. The final training loss is not shown.

**Questions:**

Refer Weakness.

---

> ### Author Response · Authors · 2023-11-20
> **Authors' response to Reviewer FS69**
>
> We thank the reviewer for viewing this paper.
> In terms of the novelty of our motivations and methodology, we would like to emphasize the key factors that might be ignored by the reviewer:
> - On the general ZSL problem, we summarize the two aspects, i.e., context diversity addressed by modeling class distribution over soft prompts and the informativeness inherited from LLM-based language prompts, are only independently studied in the literature. In this paper, we advocate that they can be naturally bridged together to take the merits of both while overcoming their limitations. This points to a brand-new prompt learning paradigm, that is general to any vision-language foundation models on zero-shot classification problems.
> - Specific to the CZSL task, we are the first to model the class-wise distribution over both the compositional and primitive classes. Besides, our VLPD is different from existing CZSL literature as we decompose both image and text modality rather than simply decomposing text features in the DFSP and HPL (see text below Eq.(2) and Table 5).
>
>
> For the technical questions:
>  - Our augmentation to image input is NOT a test-time adaptation, as its relevant module VFE has to be learned in training.
>  - For the Eq. (1), the matrix $A$ is defined to represent pair-wise class correlation on each of $d$ features so that its shape is $d\times C\times C$ where $C$ is the number of classes. Therefore, the $A_{k,y}$ is a $d$-dimensional vector. Note in practice, before the marginal logit $v^\top A_{k,y} {v}$ is computed, $A_{k,y}$ is diagonalized to have the shape $d\times d$. To avoid misunderstanding, we will update the marginal logit as $v^\top \texttt{diag}(A_{k,y}) v$ in the main paper.
>  - As requested, results w.r.t. the value of $N=0$ are shown in the Table below. It clearly shows the effectiveness of the VFE that takes an image with $N$ augmented views as input.
>  - The final training loss is a weighted sum of $\mathcal{L}_y$ in Eq. (1) and $\mathcal{L}_s+\mathcal{L}_o$ in Eq. (2). The weights between them for different datasets are summarized in Table 7 of the supplement.
>
> | Models      | $\text{H}_\text{cw}$ | $\text{AUC}_\text{cw}$ | $\text{H}_\text{ow}$  |  $\text{AUC}_\text{ow}$  |
> | :---        |    :----:   |       :----:   |       :----:   |   :----: |
> | $N=0$ |  37.89 | 21.07|19.37  | 6.78 |
> | $N=8$ | $\textbf{38.97}$     | $\textbf{22.12}$       | $\textbf{20.41}$ | $\textbf{7.34}$  |

---

### Meta-Review · Area_Chair_xYCY · 2023-12-11

**Metareview:**

The manuscript introduces an innovative approach to compositional zero-shot learning (CZSL), aiming to identify combinations of attributes and objects based on a limited subset available during training. The reviewers generally appreciate the fundamental concepts and principles outlined in the paper, with one reviewer particularly expressing a strong interest in the work. However, there are significant concerns raised by the reviewers, particularly regarding the marginal contributions of the various modules proposed in the study and the influence of Large Language Models (LLMs) in this context. Given these reservations, the area chair is currently inclined towards recommending rejection of the submission. The concerns primarily revolve around the need for a clearer demonstration of the unique contributions and the real-world impact of the proposed methodology in the field of CZSL.

**Justification For Why Not Higher Score:**

There are significant concerns raised by the reviewers, particularly regarding the marginal contributions of the various modules proposed in the study and the influence of Large Language Models (LLMs) in this context. Given these reservations, the area chair is currently inclined towards recommending rejection of the submission.

**Justification For Why Not Lower Score:**

N/A

---

### Decision · Program_Chairs · 2024-01-16

Reject